# S100A6 Protein—Expression and Function in Norm and Pathology

**DOI:** 10.3390/ijms24021341

**Published:** 2023-01-10

**Authors:** Wiesława Leśniak, Anna Filipek

**Affiliations:** Laboratory of Calcium Binding Proteins, Nencki Institute of Experimental Biology, Polish Academy of Sciences, 02-093 Warsaw, Poland

**Keywords:** S100A6 (calcyclin), expression, intracellular/extracellular role, marker, stem/progenitor cells, body fluids, tumors, pathology

## Abstract

S100A6, also known as calcyclin, is a calcium-binding protein belonging to the S100 protein family. It was first identified and purified more than 30 years ago. Initial structural studies, focused mostly on the mode and affinity of Ca^2+^ binding and resolution of the resultant conformational changes, were soon complemented by research on its expression, localization and identification of binding partners. With time, the use of biophysical methods helped to resolve the structure and versatility of S100A6 complexes with some of its ligands. Meanwhile, it became clear that S100A6 expression was altered in various pathological states and correlated with the stage/progression of many diseases, including cancers, indicative of its important, and possibly causative, role in some of these diseases. This, in turn, prompted researchers to look for the mechanism of S100A6 action and to identify the intermediary signaling pathways and effectors. After all these years, our knowledge on various aspects of S100A6 biology is robust but still incomplete. The list of S100A6 ligands is growing all the time, as is our understanding of the physiological importance of these interactions. The present review summarizes available data concerning S100A6 expression/localization, interaction with intracellular and extracellular targets, involvement in Ca^2+^-dependent cellular processes and association with various pathologies.

## 1. Introduction

The S100 protein family, of which S100A6 is a member, groups more than 20 low-molecular-weight Ca^2+^-binding proteins [1]. Most of the genes encoding S100 proteins form a cluster on human chromosome 1 and mice chromosome 3 [2,3]. These genes and their protein products are designated with letter A and a number (S100A1–S100A16) that reflects the position of a given gene in the cluster on the human chromosome. Single S100 genes located on other chromosomes are designated by other letters (e.g., S100B, S100Z, S100G and S100P).

S100A6 was originally isolated from Ehrlich ascites tumor cells [4]. It is present in numerous cell types but is particularly abundant in epithelial cells and fibroblasts [5]. S100A6 is localized mainly in the cytoplasm but is also present in the cell nucleus and readily associates with the plasma membrane and nuclear envelope when the intracellular Ca^2+^ concentration increases [6,7,8].

S100A6 is composed of 89 amino acids (mouse and rat), 90 amino acids (man and rabbit) or 91–92 residues (chicken; two isoforms) [9]. Its sequence comprises two helix–loop–helix EF-hand structures, separated by a linker region. Amino acids located in the loop region coordinate Ca^2+^. The C-terminal EF-hand, comprising helices III and IV, has a canonical Ca^2+^ binding loop containing 12 amino acids and binds Ca^2+^ with high affinity (in a 10^−6^ M range), while the N-terminal loop, between helices I and II, has 14 amino acids and its Ca^2+^- binding affinity is slightly lower (Figure 1) [10].

In the cell and in solution, S100A6 exists as a non-covalent homodimer held together by hydrophobic interactions involving helices I and IV of each monomer [10,11]. Of note, it may also form a covalent dimer through a disulfide bridge between cysteine residues at position 2, and a heterodimer with another S100 family member, S100B [12,13]. Binding of two Ca^2+^ via the EF-hand structures induces conformational changes, mainly in helices II and III, which result in exposition of a hydrophobic surface responsible for interaction with target proteins and transduction of Ca^2+^ signals [10,14]. S100A6 also binds Zn^2+^, but the resulting conformational changes are different from those observed after binding of Ca^2+^ [15,16].

S100A6 Is involved in many processes inside and outside the cell, but its primary function seems to be associated with cell proliferation/tumorigenesis, cell differentiation, cytoskeletal dynamics or cellular stress response. Present extracellularly in the matrix or different body fluids, S100A6 may serve as a diagnostic marker in some diseases. This review summarizes the current state of knowledge regarding various aspects of S100A6 biology.

## 2. Localization and Expression of S100A6

S100A6 is a ubiquitous protein present in most human tissues and organs (https://www.proteinatlas.org/ (accessed on 5 September 2022)). The highest level of S100A6 expression is found in fibroblasts and epithelial cells [5], but it is also expressed in highly specialized cells such as certain neurons [17], glial cells [18], smooth or cardiac muscle cells [19,20] or various hematopoietic cells [21,22]. Interestingly, S100A6 has been detected in adult stem/progenitor cells of various lineages, e.g., in hematopoietic, neural and epidermal stem cells [23]. Single-cell RNA-seq experiments confirmed that S100A6 expression strongly correlated with the expression of established adult stem cell markers and was, in the majority of cases, characteristic for the most quiescent stem cell subpopulation.

S100A6 is predominantly a cytoplasmic protein, but in the presence of Ca^2+^ it associates with the plasma membrane and the nuclear envelope [6,7]. There are also several reports on the presence of S100A6 in cell nuclei, for example, in smooth muscle cells [19], lung carcinoma cells [24,25], skin tumor cells [26] and pancreatic ductal adenocarcinoma cells [8].

S100A6 expression can be up-regulated by multiple factors, such as epidermal growth factor (EGF) platelet-derived growth factor (PDGF), epidermal growth factor (EGF), serum [27], tumor necrosis factor (TNFα) [28], retinoic acid [29], estrogen [30], palmitate [31], glucose [32], vasopressin [33], gastrin [34] and various cytokines [35]. Transcription of the S100A6 gene is under the regulation of the following transcription factors: USF [36], NF-κB [37], SP1 and p53 [38], p63 [39], β-catenin [40], c-myc [32], ChREBP [41], ZEB1 and STAT3 [35]. Several recent studies revealed the role of microRNAs (miRNAs) in the regulation of S100A6 expression. MiR-193a and miR-493-5p were shown to inhibit S100A6 expression by binding directly to the 3′-UTR of S100A6 mRNA [42,43], while several other miRNAs were shown to affect S100A6 level indirectly [44,45]. S100A6 expression is also controlled by DNA methylation of the S100A6 gene [46,47].

## 3. Intracellular S100A6 Ligands

Over the years, numerous protein ligands of S100A6 have been identified. Corresponding to the distribution of S100A6 described above, the list includes cytoplasmic, membrane, nuclear and extracellular proteins. It should be emphasized that in in vitro conditions the binding occurs in the presence of Ca^2+^ and is abolished when Ca^2+^ is removed by chelators. This indicates that, in the cell, the interaction between S100A6 and the ligand can be triggered by an increase in intracellular Ca^2+^ concentration due to different stimuli. Thus, S100A6 and other S100 protein family members, constitute an important link in the transmission of the Ca^2+^ signal to effector proteins.

Data on S100A6 ligand binding have been obtained by various methods, and the strength and functional significance of these interactions have not always been investigated. Nonetheless, the apparent diversity of S100A6 ligands indicates that they might be implicated in a wide scope of cellular processes. There are several processes/phenomena in which the involvement of S100A6 seems to be particularly well documented. Worthy of note, for example, is the number of cytoskeletal proteins, both structural and regulatory ones that interact with S100A6. The list contains tropomyosin [48], caldesmon [49,50,51,52,53], calponin [54,55], non-muscle myosin IIA [56], actin [57] and cofilin-1 [58]. S100A6 also binds to components of the microtubule network such as α and β tubulin [59], kinesin light chain (KLC) [60] and centrosomal proteins FOR20, FOP and OFD1 [61]. Recent work has shown that S100A6 also interacts with tubulin polymerization-promoting protein (TPPP) [62]. This list of proteins strongly implicates S100A6 in the functioning of both the actin and the tubular cytoskeleton. In particular, through interacting directly with actin and myosin, or with actin- and myosin-binding proteins, S100A6 may affect actin–myosin interactions and myosin ATPase activity [48] or interfere with multiple activities of the actin cytoskeleton [57,58]. Likewise, via interaction with tubulin, centrosomal proteins, TPPP and KLC, it may regulate tubular network organization and/or tubular transport.

Interestingly, S100A6 also interacts with lamin B1 [63] and lamin A/C [40], the building blocks of nuclear filaments, i.e., structures that maintain the structural stability of the nucleus and play a role in chromatin organization [64]. Although the functional consequences of this interaction have not been studied, it implicates S100A6 in the structural aspects of the nucleus and, potentially, in chromatin dynamics. This assumption is reinforced by a recent report identifying high mobility group protein 20A (HMG20A), a component of the lysine-specific demethylase 1(LSD1)-REST co-repressor 1 (CoREST) complex, among S100A6 ligands [65]. Another nuclear target of S100A6, importin α, implicates it in regulation of nuclear transport. Namely, the interaction of S100A6 with the Arm domain of importin α disrupts the binding between importin and the nuclear localization signal (NLS) motif in cargo proteins and results in less efficient transport to the nucleus [66]. Last but not least, S100A6 can interact with ribosomal S6 kinase 1 (RSK1) which, when phosphorylated and activated by ERK1/2, enters the nucleus via importin-dependent transport and phosphorylates/activates numerous nuclear substrates, including transcription factors [67]. Although the consequences of the S100A6–RSK1 interaction have not been determined, it is worth stressing that RSK1, like S100A6, is linked with cell proliferation, as it promotes the G1 to S transition and DNA synthesis [67].

Other targets of S100A6 are annexins (Anx), Ca^2+^ and phospholipid binding proteins involved in membrane-linked processes such as membrane aggregation, fusion or permeability [68]. Among the 13 known annexins, S100A6 was found to interact with Anx1, Anx2, Anx6 and Anx11 [69,70,71,72]. The binding involves a short helical N-terminal segment, which is the most divergent part of the annexin molecule. Since annexins play a role in the anchoring of actin filaments to the cell membrane, they can be considered as cytoskeleton-linked S100A6 ligands. Taking into account the properties of annexins, it is believed that interaction with S100A6 and other S100 proteins might be important for endocytosis, secretion and other membrane-associated cellular processes.

Another interesting group of S100A6-interacting partners consists of proteins that are part of the Hsp70/Hsp90 machinery and often possess chaperone or co-chaperone properties. To this group belong Sgt1 [73,74], CacyBP/SIP [75], melusin [76], Hop (Hsp90/Hsp70-organizing protein), KLC and Tom70 [60], CHIP (C-terminus of Hsc70-interacting protein) [77], the immunophilins FKBP38 [78], FKBP52 and CyP-40 [79], and PP5 phosphatase [80]. It is worth noting that two of those proteins, Tom70 and FKBP38, are localized in the outer mitochondrial membrane. All these ligands implicate S100A6 in cellular response to stressful conditions. Although the biological effect of these interactions is not always clear, in most studied cases the binding of S100A6 results in dissociation of the pre-existing complexes of the above-mentioned co-chaperones with Hsp90 or Hsp70, allowing for complex remodeling upon the increase in intracellular Ca^2+^ concentration. In the case of CacyBP/SIP, it was shown that S100A6 may compete for the binding with ERK1/2 [81]. Additionally, S100A6 was found to inhibit CacyBP/SIP phosphorylation by casein kinase II [82].

Yet another set of S100A6 ligands, shared with some other S100 proteins, includes transcription factors of the p53 family, namely p53, p63 and p73 [83,84,85] and ubiquitin ligase Mdm2 (mouse double minute 2) [86]. Similarly to other S100 proteins, S100A6 binds to the tetramerization domains of p53, p63 and p73 in their monomeric, dimeric and also (in the case of p63 and p73) tetrameric form [84,87]. Interestingly, it can also interact with the N-terminal transactivation domain of these transcription factors [84]. This binding, in turn, interferes with p53 interaction with Mdm2 [86] and p300 acetyltransferase [85]. The picture is further complicated by the fact that S100A6 can also interact with Mdm2, a ubiquitin ligase degrading p53 [86]. Both positive and negative effects of S100A6 on p53 activity were reported, but the net physiological outcome of the numerous interactions is difficult to estimate also due to functional redundancy in the S100 protein family.

Cell surface receptors constitute another interesting group of S100A6 ligands, especially because, as mentioned earlier, S100A6 can be released from the cell and exert its effect extracellularly. The best-studied receptor ligand is RAGE (receptor for advanced glycation end products), which belongs to the PRRs (pattern recognition receptors) and transduces signals delivered by diverse ligands, including other S100 proteins, into pro-inflammatory responses [88]. RAGE activity has been implicated in different pathologies including diabetes, neurodegenerative diseases and cancers [88]. Binding of S100A6 to the RAGE ectodomain induces its dimerization and facilitates signal transduction [89,90]. Integrin β1 is another receptor ligand of S100A6, and the binding leads to activation of GSK3β, focal adhesion kinase (FAK) and p21-activated kinase (PAK)-dependent pathways [59]. Other membrane proteins that interact with S100A6 include Na^+^/Ca^2+^ exchanger (NCX1) and the TRPM4 transient receptor potential cation channel, both involved in ion transport [56].

S100A6 was also shown to bind to soluble extracellular proteins. The list comprises cytokines such as IFN-β, IL-11, CNTF or erythropoietin [91,92,93]. Although an in vitro viability assay showed no influence of S100A6 on IFN-β-induced toxicity, this does not exclude the possibility that S100A6 and other S100 proteins can modulate cytokine signaling in vivo [93]. Other extracellular targets of S100A6 include lysozyme, lumican, prolargin (PRELP) and insulin-like growth factor binding protein 1 (IGFBP-1) [94,95]. Of note, lysozyme has been recently shown to block S100A6 binding to RAGE [95]. Lysozyme, together with FKBP38, FKBP52, Cyp-40, PP5, CHIP, RSK1 and GAPDH [96], belongs to a group of S100A6 ligands endowed with enzymatic activity. However, except for PP5 [80], there are no data on how S100A6 binding may affect their activity.

## 4. Structural Aspects of S100A6—Ligand Interactions

The minimal regions responsible for the interaction with S100A6 have been identified for many ligands. Generally, as for other S100 proteins [97], these are short, about 15–20 amino acid long protein segments with no apparent sequence similarity but usually enriched in hydrophobic and basic amino acids. These segments adopt a helical structure of amphipathic character upon binding to S100 proteins [98,99]. However, S100A6 was shown to interact also with protein motifs with defined secondary structures, such as tetratricopeptide repeat (TPR) domains. TPR is a 34 amino acid sequence forming a helix–loop–helix structure. TPRs are found in numerous S100A6 ligands, namely, Hop, KLC, Tom70, Cyp-40, FKBP38, FKBP52 and PP5 [60,77,78,79,80]. A structurally similar motif, called armadillo motif, is present in importin α and mediates its interaction with S100A6 [66]. A slightly different type of domain, which also has a helical structure, a so-called SGS domain, mediates the interaction of Sgt1 and CacyBP/SIP with S100A6 [73,100].

The S100 proteins usually interact with their ligands with a 1:1 stoichiometry, which means that two ligand molecules bind per S100A6 dimer, forming a heterotetramer. Nonetheless, binding of one ligand molecule to the S100A6 dimer was postulated for CacyBP/SIP, FOR20 and HMG20A [61,65,99]. Moreover, based on structural data, the possibility of simultaneous binding of more than one ligand cannot be excluded [10]. Numerous studies performed using biophysical methods established that, basically, the conformation of S100 proteins does not change upon ligand binding [101]. It is thus interesting how, having a rather rigid backbone, S100A6 and other S100 proteins can accommodate the binding of so many structurally divergent proteins. The canonical structure of S100 protein–ligand complexes assumes that the interacting domain of the ligand is positioned in a cleft formed, in a Ca^2+^- bound state, by the linker region between the two EF-hands and by helices III and IV of the C-terminal EF-hand [101]. In this configuration, each of the two ligand molecules would interact with a separate S100A6 monomer. However, recent methodological and analytical advances added some more complexity to this general scheme. Thus, in the case of annexin binding, it is believed, by analogy with other S100 proteins, that one end of the interacting segment of an annexin molecule contacts helix I of one S100A6 monomer while the other end interacts with residues in the linker and helix IV of the second monomer, bridging the two monomers together [101]. Yet another interaction mode was described for the complex of S100A6 with a 30 amino acid long (residues 189–219) CacyBP/SIP fragment, which is longer than most of the other binding domains studied and comprises two α helices [99]. One of those helices was shown to be positioned in a groove between helix III and IV of one S100A6 monomer, i.e., similar similarly to the canonical mode but without interaction with the linker, while the other contacted helix I of the other subunit [99]. In the complex of S100A6 mutant (C3S) with the V-domain of RAGE, the ligand interacts with amino acid residues in both Ca^2+^ binding loops and in helix III of S100A6 [89]. However, a different contact pattern and two types of complexes were proposed on the basis of the crystal structure of the complex formed by S100A6 and a RAGE fragment comprising the V, C1 and C2 domains [90]. A weak complex, in which each S100A6 monomer contacted one of the V domains through the N-terminal EF-hand domain, could be observed. In the second type of complex, each S100A6 monomer interacted with the C1 part of the RAGE fragment and simultaneously with the C2 domain of the other, thus bringing the two interacting molecules closer together [90]. This strong interaction could entail clustering of RAGE molecules and facilitate signal transduction [90]. Interestingly, such binding mode, which engages amino acid residues located both in the N-terminal and C-terminal part of S100A6, would require a different conformation of the S100A6 dimer which, as mentioned above, is considered to be stable; therefore, the existence of such a complex should be further confirmed.

### 4.1. Intracellular S100A6—Involvement in Cell Proliferation and Differentiation

S100A6 mRNA was first identified in growth-stimulated quiescent cells [102] and this finding was followed by many others, pointing to a tight correlation between high S100A6 level and cell proliferation and motility [103]. In cancer cells, this correlation extends to tumor growth and invasiveness. Data concerning S100A6 knockdown are less numerous but equally convincing. For example, it has been shown that S100A6 deficiency profoundly inhibited proliferation of fibroblasts [104,105], osteoblasts [106] and pancreatic carcinoma cells [107]. S100A6-deficient fibroblasts exhibited a prolonged G0/G1 phase of the cell cycle and demonstrated features of cellular senescence such as changes in cell shape and morphology, substantiated by altered organization of tropomyosin-containing [104] and actin filament networks [105]. S100A6 also exerts a profound impact on cell adhesion [108,109] and motile properties [63,105,110], although these may depend on cell type. Moreover, S100A6 knockdown entailed the up-regulation of several proteins recognized as negative regulators of cell division, while those involved in cell proliferation were down-regulated [107].

How the presence of S100A6 translates into cell proliferation and pro-survival cues is a matter of study (Figure 2). At least three reports show that overexpression of S100A6 results in an increase in β-catenin level or in its translocation to the nucleus [111,112,113]. β-catenin is a key mediator of the canonical Wnt signaling pathway, taking part in transcriptional regulation/stimulation of many genes, including those involved in pro-survival pathways and carcinogenesis [114]. Since β-catenin enhances S100A6 expression [40], the resulting positive feedback loop may drive cell proliferation. Another possible mechanism through which S100A6 level may influence cell proliferation involves activation of the MAP kinase signaling pathway. S100A6 overexpression was shown to result in increased phosphorylation (activation) of p38 and ERK1/2 kinases in colorectal cancer cell lines, while S100A6 knockdown corresponded to lower phosphorylation [115]. Increased p38, but not ERK1/2, phosphorylation was also detected after S100A6 overexpression in nasopharyngeal carcinoma cells, while an adverse effect was observed after S100A6 silencing [116]. S100A6 was also found to participate in the activation of yet another signaling pathway, namely the PI3K/AKT pathway [117]. Finally, as mentioned earlier, S100A6 directly interacts with RSK1 [56]. Thus, experimental data link S100A6 with several pro-survival signaling pathways, although we still do not know at which point the signal from S100A6 is integrated. In this regard, recent studies revealed that, in keratinocytes, S100A6 may act by activating the epidermal growth factor receptor (EGFR), which transmits the signal to MAPK and other kinases [118]. In addition, it was shown that S100A6, through its interaction with integrin β1, activated the following downstream kinases: GSK3β, FAK (focal adhesion kinase) and PAK (p21-activated kinase) [59]. FAK and PAK are involved in the regulation of cytoskeletal dynamics, cell motility, cell proliferation and transformation, and may act via phosphorylating MAP kinases [119]. It is also worth mentioning that extracellular S100A6, acting through RAGE, activated JNK kinase [120]. Interestingly, as revealed by microarray analysis, S100A6 not only promotes pro-survival signaling but can also attenuate anti-proliferative pathways [121].

There are much less experimental data concerning the impact of S100A6 on cell differentiation. It was shown, for example, that S100A6 overexpression delayed the appearance of epidermal differentiation markers both in classic and organotypic cultures of HaCaT keratinocytes [39]. On the other hand, S100A6 knockdown had no, or only a modest, effect on the pace of keratinocyte differentiation. Another study showed that S100A6 knockdown favored osteogenic differentiation of mouse embryonic fibroblasts, as could be judged by increased activity of alkaline phosphatase but was not sufficient to induce ectopic bone formation [122]. However, when differentiation was stimulated by the addition of BMP9, one of the most potent osteogenic factors, S100A6 knockdown cells, produced significantly more bone mass compared to control cells. S100A6 overexpression could not counteract BMP9-induced cell differentiation. These results suggest that S100A6 disturbs and/or delays cell differentiation by providing pro-proliferative cues rather than inhibiting those involved in differentiation.

### 4.2. Intracellular S100A6—Involvement in Cellular Stress Response

There are many reports showing that S100A6 level increases under stress conditions such as ischemia [123], mechanical force [124], irradiation [24], oxidative stress [125], hypertension [126] or kainic acid treatment [127]. All these conditions may lead to cell apoptosis and inflammation followed, depending on the extent of tissue damage, by regeneration [128]. Interestingly, literature data, although sometimes contradictory, indicate that S100A6 may be involved in most of the processes constituting cellular response to stress. For example, S100A6 has been shown to promote apoptosis in cells exposed to oxidative stress [83] or Ca^2+^ ionophore [129], while its decreased level made cells more resistant. In neuroblastoma cells, binding of extracellular S100A6 to RAGE induced apoptosis via/through reactive oxygen species (ROS)-dependent activation of JNK and caspases 3 and 7 [120]. An active role in stimulating caspase 3 transcription has also been postulated [129]. The pro-apoptotic effect of S100A6 may also stem from the fact that it inhibits the interaction between FKBP38 and Bcl-2; binding of FKBP38 to Bcl-2 exacerbates the anti-apoptotic properties of Bcl-2 [78]. However, there are also reports documenting anti-apoptotic effects of S100A6. For example, it was shown that S100A6 inhibits cardiac myocyte or renal carcinoma cell apoptosis [28,130].

S100A6 is often detected at sites of inflammation [131], but there are contradictory reports as to its role in this process [132,133]. On the other hand, numerous data indicate that S100A6 may be involved in tissue/cell repair and regeneration after stress. First of all, as mentioned above, it interacts with many components of the heat shock machinery. Structural studies have demonstrated that the interaction of co-chaperone proteins with S100A6 engages the same domain, namely the TPR domain, through which the co-chaperones interact with Hsp90 [60,79,80]. Accordingly, it was shown using recombinant proteins and cell lysates that S100A6 competes with Hsp70/Hsp90 for binding with co-chaperone proteins, e.g., HOP, KLC and FKBP38 [60,78]. Thus, evidently, upon stressful conditions, S100A6 can induce a rearrangement of the preexisting chaperone–co-chaperone complexes but whether this contributes to more efficient protein refolding and/or disaggregation is not known. Although S100A6 was reported to inhibit β-amyloid aggregation [134] or favor β-amyloid plaque disaggregation [135], the effect was ascribed to direct interaction with β-amyloid or to Zn^2+^ chelation, respectively, rather than to its interaction with the components of the chaperone system. Other data link S100A6 with tissue regeneration due to its pro-proliferative properties. For example, an increase in S100A6 level after injury was interpreted as beneficial in the case of renal tubular cells or the spinal cord [136,137], since it correlated with cell proliferation or neurogenesis. Likewise, an increased S100A6 level accompanied hair and hematopoietic stem cell regeneration [138,139]. In accordance with this are the results showing that expression of S100A6 in various brain structures is different in mice subjected to mild chronic stress when compared to control mice [140]. A significant decrease in the protein level of S100A6 was observed in brainstem structures and also in the olfactory bulb, cerebellum and stress-related structures such as the hippocampus and the hypothalamus of stressed animals. An initial decrease in S100A6, followed by a gradual increase up to control levels, was detected in the hippocampus of rats subjected to traumatic brain injury [141]. These observations suggest that S100A6 expression in the brain is affected by stressful conditions to which the animals are exposed.

## 5. Extracellular S100A6—A Marker of Pathological States

S100A6, like some other S100 family members, can be secreted from cells most likely by a microtubule-dependent mechanism [59], and can be found in cell culture media and various physiological fluids. The presence of S100A6 has been reported for example in the culture medium of decidual cells [142] or cerebellar granule neurons treated with myelin-associated glycoprotein [143]. It has been also detected in extracellular matrix of Wharton’s jelly [144], amnion fluid [145], pancreatic juice [107,146], urine [147,148] and tears [149]. Recent results have shown that S100A6 is present in nasal lavage fluid [150], and its level is elevated in the nasal swabs of COVID-19 patients [151]. The role of extracellular S100A6 has not been extensively studied. However, it was proposed to regulate secretory processes since, when added to the medium, it stimulated the secretion of lactogen II by trophoblast cells [142] and the release of insulin from pancreatic cells [41,152] and of histamine from mast cells [153]. In the case of mesenchymal stem cells of the Wharton’s jelly extracellular S100A6 caused an increase in cell adhesion and, contrary to the intracellular protein, inhibited their proliferation [144].

There are numerous reports linking changes in S100A6 level in body fluids with various diseases. For example, serum levels of S100A6 are positively correlated with the progress of four different types of cancer: gastric cancer, non-small cell lung cancer, ovarian cancer and urinary bladder cancer [42,148,154,155,156,157,158,159]. Based on these data, it has been proposed that the serum level of S100A6 can be used as a biomarker in detecting these types of cancer and serve as an indicator of their progress. As for other diseases, an increase in the serum level of S100A6 has been observed in acute coronary syndrome [160] and primary biliary cholangitis [161]. Interestingly, the serum concentration of S100A6 was diminished in the second trimester of pregnancy in women suffering from preeclampsia [162]. In the systemic sclerosis of the lung, an increased concentration of S100A6 has been detected in the bronchoalveolar lavage fluid [163,164]. S100A6 level in tears increases in various eye pathologies [149,165,166]. A higher level of S100A6 has also been found in the amniotic fluid following intra-amniotic infection [167], and in the vaginal fluid where it likely exerts an anti-microbial function [168]. Interestingly, decreased levels of S100A6 have been observed in the sputum of people living in a high background radiation area. This might be correlated with their adaptive response to the environment, which is also reflected in their low mortality from cancer [169].

An increased/diminished level of S100A6 present in body fluids is probably a secondary phenomenon that reflects changes in cellular S100A6 expression and thus may serve as an excellent marker of various pathological states. However, extracellular S100A6 may also contribute to the severity of a disease by activating certain intracellular signaling pathways through binding to cell receptors, as shown in Figure 2.

## 6. S100A6 in Various Pathologies

An increased level of S100A6 has been observed in different pathologies, including cancers. Although it is not known whether enhanced S100A6 expression is the cause or effect of pathological changes, it is rather certain that S100A6 has its share in the severity and progress of many diseases, since its knockdown in cellular or animal disease models usually alleviates the symptoms.

Among non-cancer pathologies, a high S100A6 level is typical for fibrosis-related diseases, characterized by excessive deposition of extracellular matrix proteins by activated fibroblasts [170]. As mentioned above, fibroblasts are particularly rich in S100A6 [5]. A high S100A6 mRNA/protein level was found in liver cirrhosis biliaris [5,171,172], chronic renal disease [173], pulmonary fibrosis [163] and myocardial infraction [174]. Recently, S100A6 expression has been shown to be highly correlated with CCl4-induced liver fibrosis in mice. Recombinant human S100A6 introduced to these mice enhanced the symptoms. Furthermore, S100A6 induced cell cycle transition from the S to G2 stage and significantly elevated the level of ERK1/2 phosphorylation. In contrast to S100A6, the soluble receptor for advanced glycation end products (sRAGE), a natural antagonist of the S100A6/RAGE pathway, had a preventative effect on liver fibrosis in the same mouse model [175].

High/increased S100A6 level may also be an indication of astrogliosis, a pathological change observed in neurodegenerative diseases such as Alzheimer disease [134,176,177] or amyotrophic lateral sclerosis (ALS) [178,179,180,181]. Interestingly, S100A6 localizes to β-amyloid plaques [134]. An increased level of S100A6 was also found in the hippocampus and prefrontal cortex of patients with postoperative neurocognitive disorders (po-NCD) [182]. A recent study has identified differentially expressed S100A6 in the spinal cord after injury [137]. In preeclamptic patients, the level of S100A6 in the umbilical cord was found to be much higher than in healthy pregnant women. In addition, post-translational modifications of S100A6 and its protein ligands are different in diseased women than in healthy ones [183]. Significantly higher levels of mRNA and protein of S100A6 are also characteristic for endometriosis, a benign growth of endometrial tissue outside the uterus. S100A6 expression is enhanced in ectopic endometrial tissues compared to eutopic ones [184]. S100A6 knockdown in ectopic endometrial stromal cells (ESCs) suppressed p38/MAPK activity and inhibited cell viability, migration and invasion, suggesting that S100A6 may contribute to the pathogenesis of endometriosis.

Enhanced S100A6 expression has been detected in most cancer tissues with a possible exception of testis cancer [185] (https://www.proteinatlas.org/ (accessed on 5 September 2022)). Most importantly, in many studied cases, a positive correlation between S100A6 expression and the disease stage, tumor size and/or metastasis has been reported. This applies, for example, to melanoma [186], adenoma and adenocarcinoma specimens [187] and premalignant and malignant pancreatic ductal cells [8]. In stomach cancer, S100A6 overexpression was associated with larger tumor size and deeper invasion [188]. In addition, the level of S100A6 correlated with tumor metastasis in osteosarcoma [110] and prostate cancer [46]. Regarding osteosarcoma, the latest study has shown that S100A6 overexpression increased the proliferation and reduced the osteogenic differentiation of osteosarcoma cells [43]. S100A6 expression was also helpful in discriminating between various cancer types, as in the case of cholangiocarcinoma and hepatocellular carcinoma [189,190,191]. Differences in S100A6 level proved to also be useful for the discrimination between primary liver tumors such as hepatocellular carcinoma and metastases derived from colorectal carcinoma [192,193]. All these data, together with the fact that S100A6 is detected in cancer stem cells [23], point to a potential role of S100A6 in the development of malignancy. In addition, due to changes in S100A6 expression during cancer progression, the protein has been recognized as a useful diagnostic and prognostic tool for defining cancer stage and patient prognosis as, for example, in the case of pancreatic and lung adenocarcinoma [194,195].

## 7. Conclusions

S100 proteins, due to their ability to bind Ca^2+^, serve as “readers” of Ca^2+^-conveyed intracellular and extracellular signals, which they subsequently translate into protein–protein interactions. S100A6 shares many structural features with other members of the S100 protein family but, at the same time, possesses a specific spectrum of ligands (listed in Table 1) and, possibly, functions. Despite years that have passed since its discovery the protein is still being intensively studied, as each novel ligand identified or a newly demonstrated link to some pathology open new research perspectives. Most actual topics of research include the mechanism through which S100A6 contributes to the activation of various kinases engaged in pro-survival pathways, its role in stem cells and, invariably, its role in various human pathologies, with a special emphasis on its potential use as a disease marker in clinics. Thus, S100A6 is a protein with a long history and most likely an interesting future still ahead.

## Figures and Tables

**Figure 1 ijms-24-01341-f001:**
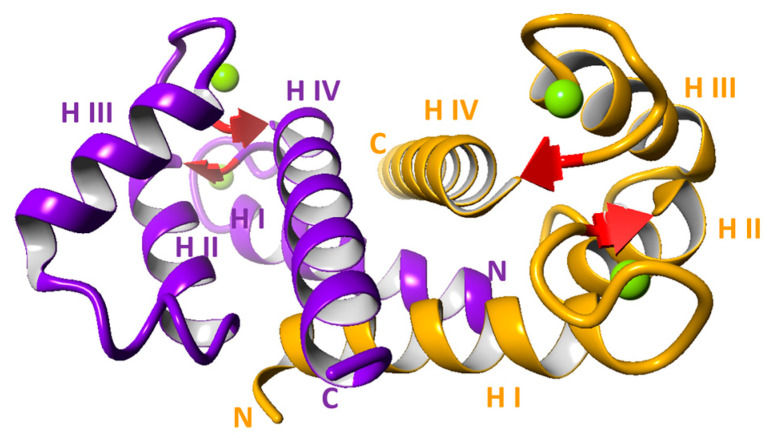
Structure of S100A6 dimer with 4 Ca^2+^ (Protein Data Bank id:1K96) [10]. Each monomer is represented by different color (violet and yellow); green balls represent Ca^2+^; H indicates helix; N and C indicate N- and C-terminus, respectively.

**Figure 2 ijms-24-01341-f002:**
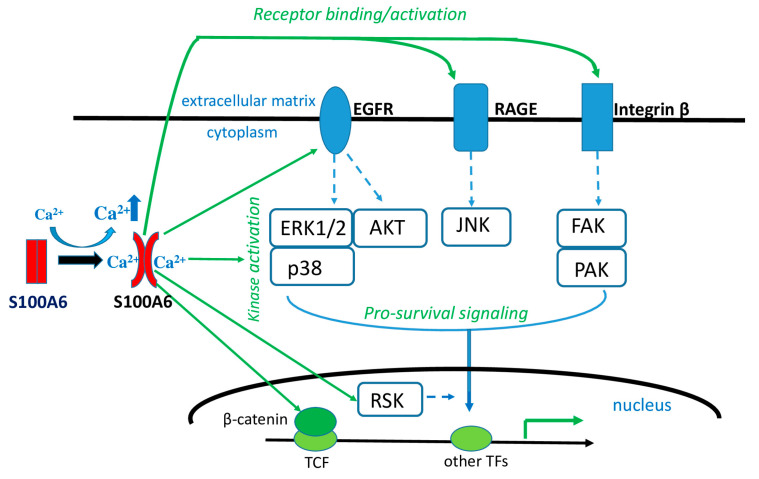
Role of intracellular and extracellular S100A6 in Ca^2+^-cellular signaling. Increased intracellular Ca^2+^ level induces a conformational change of S100A6 molecule, which enables it to interact with various intracellular proteins or, following secretion, to bind to receptor proteins and activate signaling pathways and, ultimately, gene transcription that leads to cell survival/proliferation. TF—transcription factor.

**Table 1 ijms-24-01341-t001:** Intracellular and extracellular S100A6 ligands.

Ligand	Target Region	Direct Effect	Proposed Function	Reference
Actin and tropomyosin	ND	ND	Regulation of actin filament dynamics	[48,57]
Caldesmon	C-terminus	Decrease in caldesmon affinity to actin and heavy meromyosin (HMM)	Regulation of caldesmon function in smooth muscle contraction	[49,50,51,52,53]
Calponin	ND	Stabilization of calponin	Regulation of calponin function in smooth muscle contraction	[54,55]
Coffilin	ND	Decrease in actin depolymerization	Regulation of actin polymerization	[58]
Non-muscle myosin IIA	ND	ND	Modulation of cell motility	[56]
Tubulin α and β	ND	ND	Regulation of microtubule organization	[59]
Kinesin light chain (KLC)	TPR domain	dissociation of the KLC-JIP1 complex	Modulation of KLC–cargo interaction	[60]
FOR20, FOP and OFD1	N-terminus (aa 1–30)	ND	Modulation of cilia formation	[61]
TPPP (tubulin polymerization-promoting protein)	C-terminal region (aa 110–160)	Inhibition of TPPP dimerization	Modulation of microtubule organization	[62]
Lamin B1 and A/C	ND	ND	Regulation of chromatin organization	[40,63]
Importin α	Armadillo motif; NLS-cargo-binding domain	Inhibition of importin–NPM1 binding	Inhibition of nuclear transport	[66]
HMG20A	C-terminus (aa 311–347)	ND	Regulation of neuronal differentiation	[65]
RSK1	ND	ND	Regulation of cell survival and proliferation	[56]
Annexins: Anx1, Anx2, Anx6 and Anx11	N-terminus	ND	Modulation of membrane dynamics	[70,71,72,96]
Sgt1	SGS domain	Inhibition of Sgt1-Hsp90 binding	Modulation of chaperone complexes	[74]
CacyBP/SIP	SGS domain	Inhibition of CacyBP/SIP phosphorylation by CKII	Regulation of CacyBP/SIP phosphatase activity and ERK1/2-Elk-1 signaling pathway	[81,82,100]
Hop and TOM70	TPR domain	Dissociation of Hop and TOM70 complexes with Hsp90 or Hsp70	Modulation of chaperone complexes	[60]
Melusin	C-terminus	ND	Regulation of cardioprotective pathway	[76]
CHIP	TPR and U-box domains	Moderate inhibition of CHIP interaction with Hsp90 and Hsp70; suppression of mutant p53 degradation	Modulation of chaperone complexes and p53 degradation	[77]
Cyp-40 and FKBP52	TPR domain	Inhibition of Cyp-40 and FKBP52 binding with Hsp90	Modulation of chaperone complexes	[79]
FKBP38	TPR domain	Dissociation of the FKBP38-Hsp90 and FKBP38-Bcl2 complexes; suppression of Bcl2 stability	Modulation of chaperone complexes	[78]
PP5	TPR domain	Dissociation of the PP5-Hsp90 complex; stimulation of PP5 activity	Modulation of chaperone complexes	[80]
P53, p63 and p73	P53: C-terminal tetramerization domain (aa 293–393); N-terminal transactivation domain (aa 1–57)	Inhibition of p53 tetramerization	Regulation of p53, p63 and p73 oligomerization and activity	[83,84,87]
Mdm2	N-terminus (aa 2–125)	ND	Moderate inhibition of p53 ubiquitination	[86]
RAGE	V, C1 and C2 domains	Induction of RAGE dimerization/clustering	Modulation of signal transduction through RAGE	[89,90,120]
Integrin β1	Extracellular domain	Increase in FAK and PAK phosphorylation	Modulation of integrin- dependent signaling	[59]
NCX1 and TRPM4	ND	ND	Modulation of ion transport	[56]
Erythropoietin	N- and C- terminus	ND	Regulation of erythropoietin secretion and/or activity	[92]
Cytokines, e.g., IFN-β, IL-11 and CNTF	ND	No effect on IFN-β-induced cytotoxicity	Modulation of cytokine activity/signaling	[91,93]
Lysozyme	N-terminus	Inhibition of S100A6-RAGE interaction	Modulation of RAGE-dependent signaling	[55,95]
Lumican, PRELP and IGFBP-1	ND	Competition of the IGF-1–IGFBP-1 interaction	Remodeling of extracellular matrix	[94]
GAPDH	ND	ND	Regulation of metabolic processes	[96]

TPR—tetratricopeptide; SGS—SGT1 specific, ND—not determined.

## Data Availability

Not applicable.

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
