# Peer review of "S100A6 Protein—Expression and Function in Norm and Pathology"

_ijms, 2023, doi:10.3390/ijms24021341_

Round 1

Reviewer 1 Report

This review focuses on the role of S100A6 in cellular processes and association with various pathologies. The present review is thorough and often well written. However, there are some issues to be illustrated.

1. The role of S100A6 in calcium signaling has not been clearly described. A graphic sheme would be helpful and would make this aspect easier to understand.

2. The authors should clearly summarize the dual role of S100A6 in various diseases. It is also desirable to illustrate Intracellular S100A6 and Extracellular S100A6 functions with overview figures.

Minor points

1. The formatting is not always correct in the manuscript, for example, on page 2 the font size changes

Author Response

This review focuses on the role of S100A6 in cellular processes and association with various pathologies. The present review is thorough and often well written. However, there are some issues to be illustrated.

  1. The role of S100A6 in calcium signaling has not been clearly described. A graphic scheme would be helpful and would make this aspect easier to understand.

Answer:

On page 4 of the review manuscript and in Conclusions we have clearly indicated that S100A6 can exert its biological effect (via interacting with other proteins) only when the intracellular calcium level increases. We have now added another sentence on page 4 to reinforce this message. Moreover, we have included a graphical representation of S100A6 activation by calcium in an additional figure (Figure 2).

  1. The authors should clearly summarize the dual role of S100A6 in various diseases. It is also desirable to illustrate Intracellular S100A6 and Extracellular S100A6 functions with overview figures.

Answer:

 According to the comment we have prepared a new figure (Figure 2) illustrating the intracellular and extracellular functions of S100A6 that lead to pro-survival signaling in the cell.

We are not sure whether we correctly understand the Reviewer’s comment on the dual role of S100A6 in various diseases. If it concerns the role of intracellular versus extracellular S100A6 we are inclined to think that intracellular S100A6 may have a more causative role in pathologies while the level of S100A6 present in body fluids is a secondary phenomenon that reflects the increase in cellular S100A6 expression. Nonetheless, secreted S100A6 may also contribute to the severity of a disease by activating certain intracellular Ca2+-signaling pathways through binding to cell receptors as shown in Figure 2. A sentence has been added (page 8) to clarify this point. 

Minor points

  1. The formatting is not always correct in the manuscript, for example, on page 2 the font size changes

Answer:

 This has been corrected.

Thank you for your comments that improved the quality of our manuscript.

Reviewer 2 Report

Manuscript  review written by Lesniak W and Filipek A present a literature data about S100A6 calcium binding protein belonging to S100 family of calcium binding and transporting proteins playing significant role in activation of multiple intracellular signaling pathways. Review is focused mostly on biophysical analysis of structure and versatility of S100A6 complexes interactions with various ligands. S100 A6 expression is  changed  in many different pathological  states and could be served as a potent regulator in some  diseases. Also, authors described role of S100A6 in activation of many highly  important signaling pathways such as PI3k/Akt that considering as a major signaling hub in activation of  many intracellular  protein kinases.  Manuscript is well written, some of the aspects of S100A6 expression described in details, but I didn't find ( except one sentence for COVID 19) any examples of S100A6 role in various  infectious diseases . S100A super family proteins play significant role as a pro and anti- inflammatory regulators  in  immune response to different infectious agents including viruses, microbes and parasites.  I would like to recommend to include more information about role of S100A6 as an important regulator in immune response against  of  various pathogens pathogens. 

Author Response

Manuscript  review written by Lesniak W and Filipek A present a literature data about S100A6 calcium binding protein belonging to S100 family of calcium binding and transporting proteins playing significant role in activation of multiple intracellular signaling pathways. Review is focused mostly on biophysical analysis of structure and versatility of S100A6 complexes interactions with various ligands. S100 A6 expression is changed in many different pathological states and could be served as a potent regulator in some diseases. Also, authors described role of S100A6 in activation of many highly important signaling pathways such as PI3k/Akt that considering as a major signaling hub in activation of many intracellular protein kinases.  Manuscript is well written, some of the aspects of S100A6 expression described in details, but I didn't find (except one sentence for COVID 19) any examples of S100A6 role in various infectious diseases. S100A super family proteins play significant role as a pro and anti- inflammatory regulators in immune response to different infectious agents including viruses, microbes and parasites.  I would like to recommend to include more information about role of S100A6 as an important regulator in immune response against of various pathogens pathogens. 

Answer:

We did not include this particular topic in the review because, according to the literature search we performed, there seems to be no obvious, well-documented association between S100A6 and infectious diseases. This is in sharp contrast to what is known about three other members of the S100 protein family, namely S100A8, S100A9 and S100A12, which are classified as alarmins or DAMPs (Danger-Associated Molecular Pattern) and were shown in numerous publications to play an important role in cellular defense against pathogens.

In fact, we found very few papers (eg. Zhou et al. iScience, 2021; Tani J, Liver Int., 2013) that study possible involvement of S100A6 in this kind of diseases. Illustrative in this regard is the paper by Kazakov et al., 2020 (cited by us) in which the Authors, taking advantage of 2 different databases, present the list of diseases (Tabs S6 and S7) with possible association to S100A6 (and to interferon β). According to 2 different databases there are only 3 infectious diseases that can be possibly associated with S100A6. Of note, the associative data are based on a very limited publication number and the association coefficients are low. In the same paper the Authors show a direct binding of S100A6 with interferon β, a cytokine with an established role in suppression of viral and bacterial infections; however, they also show that the binding has no effect on the biological effects of interferon β. Based on this we do not feel that we can add something to our text.

Round 2

Reviewer 1 Report

the authors have answered all the questions